# Synthesis of Yttrium Oxide Nanoneedles with Carbon Dioxide Carbonization

**DOI:** 10.3390/nano12193440

**Published:** 2022-10-01

**Authors:** Minglu Rao, Anbang Lai, Miaomiao Zan, Menglan Gao, Yanfei Xiao

**Affiliations:** 1Faculty of Materials Metallurgy and Chemistry, Jiangxi University of Science and Technology, Ganzhou 341000, China; 2Ganzhou Engineering Technology Research Center of Green Metallurgy and Process Intensification, Ganzhou 341000, China; 3Key Laboratory of Ionic Rare Earth Resources and Environment, Ganzhou 341000, China

**Keywords:** carbon dioxide, carbonization, 1D needle-like structure, crystallization mechanism, yttrium oxide

## Abstract

In this study, a CO_2_ carbonization method is introduced for the preparation of 1D yttrium oxide powders. Using YCl_3_ as the raw material, sodium hydroxide was initially used to completely precipitate Y^3+^ into yttrium hydroxide, and then CO_2_ was introduced into the yttrium hydroxide slurry for homogenization-like carbonization to obtain yttrium carbonate precipitation. Then, by studying the effects of carbonization conditions, such as the temperature, CO_2_ flow rate, and stirring speed, on the morphology and phases of yttrium carbonate, the temperature was observed to exert a greater effect than the other experimental parameters on the morphology and structure of the carbonized products. Finally, Y_2_(CO_3_)_3_·2H_2_O nanoneedles were obtained at optimal conditions. The carbonized crystals of the acicular yttrium carbonate precipitate because of the solution supersaturation and then quickly complete their crystal growth process through the oriented attachment (OA) and Ostwald ripening (OR) mechanisms. After heat treatment, yttrium carbonate retained a good crystal morphology and produced Y_2_O_3_ nanoneedles with a length of 1–2 μm and a width of 20–30 nm.

## 1. Introduction

Continuous in-depth research by scientific and technological workers has established that the physical and chemical properties of materials are affected not only by their composition and particle size but also by their morphology [1,2,3,4,5,6]. When electrons are confined in a one-dimensional (1D) quantum wire, they only have one degree of freedom along the direction of the quantum wire, which alters the scattering effect of the electron movement, resulting in the physicochemical properties of 1D materials being different from other dimensions [7,8,9]. Yttrium oxide has excellent properties, such as optical properties, catalytic capabilities, and high-temperature stability [10,11,12]. It is one of the most promising rare-earth compounds and has been widely used in laser crystals, high-brightness fluorescent substrates, catalysts, and fine ceramics [13]. Yttrium oxide with a one-dimensional (1D) structure not only has the unique optical, electrical, and magnetic properties of the rare earth element yttrium, but also has excellent characteristics of 1D structural materials, and its design and synthesis have received increasing attention [7].

To date, scientific and technological workers have made many pioneering attempts to control the synthesis of 1D nanostructured yttrium oxide materials. For example, Li successfully prepared rod-shaped, needle-shaped, and other shapes of Y_2_O_3_ with controllable morphologies and sizes by simply adjusting the hydrothermal reaction temperature and the initial pH value of the starting solution [14]. Kaszewski used a microwave-assisted hydrothermal method to prepare needle-like Y_2_O_3_:Tb nanomaterials with a length of 1–5 µm and a width of 100–300 nm [15]. Zhang used soluble yttrium nitrate as the yttrium source and triethylamine (TEA) as an alkaline reagent and complexing agent to successfully prepare single crystal yttrium hydroxide nanotubes under proper hydrothermal conditions, and Y_2_O_3_ nanotubes were obtained after calcination [16]. Chen used a solvothermal method with the use of the mixture of water and dimethylformamide (DMF) as a solvent to synthesize a series of rare earth oxides with 1D structures, such as Y_2_O_3_, Pr_2_O_3_, and Nd_2_O_3_ [17]. The 1D-structured Y_2_O_3_ material synthesized using the aforementioned hydrothermal and solvothermal methods exhibits excellent properties as a fluorescent matrix and catalytic material, but a template or surfactant must be added in advance to control the directional growth of the precursor crystal nucleus in a certain crystal plane and generate crystals that preferentially grow into the morphological structures required for practical applications. These methods use harsh reaction conditions and have relatively complex requirements for equipment and operation, high costs, and small processing volumes. Therefore, the research and development of low-cost, easy-to-operate crystal growth control technologies are urgently needed to achieve the preparation of 1D-structured yttrium oxide powders.

However, the design and synthesis of Y_2_O_3_ nanomaterials with 1D structures using the liquid phase method without templates, surfactants, and high-temperature conditions are immensely challenging, and reports from only a few related studies are available. For example, Zhang used YCl_3_ as the raw material and ammonium bicarbonate as the precipitating agent. By aging the obtained yttrium carbonate crystals for 24 h, a single needle-like yttrium oxide powder or spherical-shaped yttrium oxide powder composed of needles was synthesized [18]. However, this type of method generally requires a long time for aging; a long production cycle is not suitable for industrialized production, and the use of ammonium salts as precipitation agents is prone to problems such as ammonia nitrogen contamination. The CO_2_ carbonization method is a low-cost, simple, high-efficiency, low-pollution, and easy industrial precipitation control method. In this method, CO_2_ gas is introduced into the metal hydroxide slurry for carbonization such that CO_3_^2−^ or HCO_3_^−^ is slowly and uniformly produced in the slurry, and a carbonate precipitate is formed with metal ions, thereby obtaining the target precipitation product. Recently, scholars prepared powder materials, such as acicular calcium carbonate and magnesium carbonate, with a uniform morphology using the CO_2_ carbonization method [19,20].

Based on the discussion above, this study introduces the carbon dioxide carbonization method into the preparation process for yttrium oxide nanopowders. First, sodium hydroxide is used to completely precipitate Y^3+^ into yttrium hydroxide, and then CO_2_ is introduced into the yttrium hydroxide slurry for carbonization. The carbonization conditions, such as the reaction temperature and CO_2_ flow rate, are controlled to achieve the effect of homogenization-like carbonization. Finally, Y_2_(CO_3_)_3_·2H_2_O with a 1D nanoneedle structure was prepared, and 1D nanoneedle-structured Y_2_O_3_ was obtained after heat treatment of Y_2_(CO_3_)_3_·2H_2_O. Furthermore, in this study, X-ray diffraction (XRD), scanning electron microscopy (SEM), and transmission electron microscopy (TEM) analyses were used to study the carbonization process in detail and study the crystal growth mechanism of the nanoneedle Y_2_(CO_3_)_3_·2H_2_O crystals prepared using carbonization. The carbon dioxide carbonization method for preparing yttrium oxide nanoneedles is simple, highly efficient, easy to industrialize, and does not require templates or surfactants. This approach achieves the high-value utilization of the yttrium and provides a certain basis and theoretical guidance for controlling the synthesis of other rare-earth oxide powders.

## 2. Experimental Procedures

### 2.1. Process for Synthesizing Yttrium Oxide Nanoneedles

The process used to prepare the yttrium hydroxide slurry is described below. Starting with a certain volume of high-purity YCl_3_ feed solution (provided by Ganzhou Zhanhai New Material Technology Co., Ltd. (Ganzhou, China), purity 99.99%, pH approximately 0.7), deionized water was added to prepare 500 mL of a YCl_3_ solution with a specified concentration, and the solution was placed in a 1 L beaker. A water bath was used to control the reaction temperature, and a stirring blade was used to control the stirring speed. A specified concentration of a NaOH solution (the molar concentration of sodium hydroxide was 3.33 times the concentration of yttrium chloride) was added through a peristaltic pump at a flow rate of 11.5 mL/min, and the pH of the YCl_3_ solution was slowly adjusted to 8.4. YCl_3_ was completely precipitated into a Y(OH)_3_ slurry, and then a small amount of deionized water was added to adjust the volume to 1 L. At this time, the concentration of Y(OH)_3_ was half of that of the YCl_3_ solution.

The addition of the NaOH solution induced yttrium ion precipitation as yttrium hydroxide, and because the supersaturation of the system was high, a large amount of yttrium hydroxide nucleated, grew, and agglomerated [21]. As shown in the SEM image in Figure 1, flower-like agglomerates are formed through the agglomeration of a large number of nanosheet crystals. The XRD analysis shows that its diffraction peak is broad and weak, and it is an amorphous precipitate.

The carbonization precipitation process for carbon dioxide is described below. CO_2_ was introduced into the aforementioned yttrium hydroxide slurry at a certain flow rate for 90 min at a specific carbonization temperature and stirring speed. Samples were removed at different time points during the carbonization process, and then the carbonization products were obtained by filtering, washing, and vacuum-drying at 50 °C. The carbonization products were calcined in a 600 °C high-temperature muffle furnace for 2 h to obtain the Y_2_O_3_ powder.

### 2.2. Testing and Characterization

XRD (Bruker D8, Advance) was performed to analyze the phase composition of the sample. The samples were measured with a copper target Cu Kα as the radiation source, an incident wavelength λ = 0.15406 nm, tube voltage of 40 V, and speed of 10°/min. The scan angle range was set to a 5–90° continuous scan. SEM (MLA650F, American FEI company) and TEM (JEM-2100F, JEOL) were used to observe the microscopic morphology and agglomeration states of the sample.

## 3. Results and Discussion

### 3.1. Carbonization Reaction Process

During the experiment, a NaOH solution was added to adjust the initial pH of the YCl_3_ solution to 8.4 to ensure that yttrium chloride was directly precipitated into Y(OH)_3_, and then CO_2_ was introduced into the carbonization reaction to convert Y(OH)_3_ into yttrium carbonate. As shown in Figure 2, the carbonization reaction was mainly conducted in the gas–liquid-solid reaction system, and the entire process includes CO_2_ dissolution and absorption, dissolution of yttrium hydroxide, and crystallization of yttrium carbonate. At the beginning of the carbonization reaction, the pH of the system is between the pKa1 and pKa2 of H_2_CO_3_ (Ka1 and Ka2 are the dissociation constants of H_2_CO_3_: pK = -lgK) [22]. At this time, the introduced CO_2_ quickly reacts with excess OH^−^ in the solution to form HCO_3_^−^; HCO_3_^−^ reacts with Y^3+^ to release hydrogen ions while promoting the dissolution of Y(OH)_3_. This stage is the rapid carbonization stage, and the carbonization reaction rate is controlled by CO_2_ absorption. With the progress of carbonization, the pH of the system gradually decreases. When the pH of the system is <pKa1, the dissolution of CO_2_ gradually reaches saturation, and H_2_CO_3_ is dominant in the system. Then, H_2_CO_3_ ionizes to form HCO_3_^−^ and H^+^, and H^+^ makes the yttrium hydroxide continuously dissolves. The carbonization reaction rate is mainly controlled by the dissolution of Y(OH)_3_ [23,24]. When the product of CO_3_^2−^ produced by ionized HCO_3_^−^ and the Y^3+^ concentration is greater than the solubility level of yttrium carbonate, yttrium carbonate crystal nuclei are first generated in the liquid phase, and then Y^3+^ and CO_3_^2−^ diffuse to the surface of the crystal nucleus and completely crystallize to produce the yttrium carbonate crystal structure. The nucleation and growth of the yttrium carbonate crystals continue to consume Y^3+^ and CO_3_^2−^, which promotes the continuous dissolution of carbon dioxide and yttrium hydroxide. As long as CO_2_ is continuously introduced, yttrium hydroxide will continue to dissolve, and yttrium carbonate crystals will continue to form until the end of the carbonization precipitation process. The conditions of the carbonization reaction are controlled to achieve the effect of homogenization-like precipitation. The entire carbonization process included the following reactions [25,26,27,28]:(1)$$Y{\left(\mathrm{OH}\right)}_{3}\left(s\right)⇌Y^{3+}\left(\mathrm{aq}\right)+3{\mathrm{OH}}^{-}\left(\mathrm{aq}\right)$$
(2)$${\mathrm{CO}}_{2}\left(\mathrm{aq}\right)+{\left(\mathrm{OH}\right)}^{-}⇌{\mathrm{HCO}}_{3}^{-}\left(\mathrm{aq}\right)$$
(3)$${\mathrm{CO}}_{2}\left(g\right)+H_{2}O⇌H_{2}{\mathrm{CO}}_{3}\left(\mathrm{aq}\right)⇌H^{+}\left(\mathrm{aq}\right)+{\mathrm{HCO}}_{3}^{-}\left(\mathrm{aq}\right)$$
(4)$$2Y^{3+}\left(\mathrm{aq}\right)+3{\mathrm{HCO}}_{3}^{-}\left(\mathrm{aq}\right)+2H_{2}O\to Y_{2}{\left({\mathrm{CO}}_{3}\right)}_{3}\cdot 2H_{2}O\left(s\right)+3H^{+}\left(\mathrm{aq}\right)$$
(5)$$3H^{+}\left(\mathrm{aq}\right)+Y{\left(\mathrm{OH}\right)}_{3}\left(s\right)⇌Y^{3+}\left(\mathrm{aq}\right)+3H_{2}O\left(\mathrm{aq}\right)$$

### 3.2. Research on the Process for Synthesizing Needle-like Yttrium Carbonate Crystals

#### 3.2.1. Effects of the Carbonization Temperature on the Morphology and Phase Type of Carbonization Products

Temperature changes alter the energy difference between crystal planes, resulting in different relative growth rates of the crystal planes. The resulting morphology of the crystal changes. In addition, temperature exerts an important effect on the saturated solubility, absorption and diffusion rate of carbon dioxide, the dissolution rate of yttrium hydroxide, and the supersaturation of yttrium carbonate. Therefore, the experiment first explored the effects of the carbonization temperature on the morphology and phase type of carbonization products and performed SEM and XRD analyses on the carbonized products obtained at different temperatures. The results are shown in Figure 3 and Figure 4. Figure 3 shows the SEM images of the carbonization products. It can be clearly seen from the images that the carbonization temperature evidently exerts a marked effect on the morphology of the product. When the carbonization temperature is 20 °C, the carbonized product is a large agglomerate with different particle sizes; its morphology is more consistent with the morphology of Y(OH)_3_. This morphology may be attributed to the low temperature and slow dissolution rate of Y(OH)_3_, which requires a longer time for the carbonization reaction. At the end of the experiment, most of the Y(OH)_3_ is not carbonized by carbon dioxide. When the carbonization temperature is 40 °C, the carbonized products are needle-like aggregates, and the powder has a uniform morphology. The high-magnification SEM images show that the aspect ratio is relatively large, divergent, and stacked in a staggered manner. When the carbonization temperature increases to 60 °C, the morphology of the carbonized product changes greatly, and a two-dimensional growth trend appears, forming a two-dimensional sheet-like stacked product. As shown in the high-magnification SEM images, the sheet-like carbonized product is directionally aggregated by a small needle sheet. Upon heating to 80 °C, the carbonized products form smooth flakes with large particles. The grain formation rate equation and the Kelvin formula indicate that, when the solute content in the solution is constant, the supersaturation of the solution generally increases with decreasing temperature [29]. Therefore, when the carbonization temperature is low, more CO_2_ dissolution occurs, the high supersaturation of the system is conducive to the formation of grains, and uniform small-particle crystals readily form. When the temperature is higher, the kinetic energy of the system molecules is large, CO_2_ dissolution is reduced, and the low supersaturation of the system is conducive to the growth of grains, making the crystal faces more fully developed and facilitating the formation of large-particle crystals. Furthermore, the higher the carbonization temperature, the smaller the energy difference of each crystal face of yttrium carbonate and the easier it is for each crystal plane to grow uniformly, thereby forming a multidimensional morphology. The flocculent agglomerates shown in the red circle in Figure 3 may be yttrium hydroxide. Because the temperature was high, less CO_2_ dissolved, resulting in the same CO_2_ flow rate and carbonization time conditions, and yttrium hydroxide could not be completely converted into yttrium carbonate crystals. The product has a uniform morphology and no other phases at 40 and 60 °C.

Figure 4 shows the XRD patterns of the carbonization products. When the reaction time is 90 min, the product obtained using carbonization at 20 °C is an amorphous precipitate. We tested its chemical composition and found that its carbon content was lower than at other temperatures. After comparing the XRD patterns of yttrium hydroxide, the XRD patterns of the two samples are similar, indicating that the product is incompletely carbonized yttrium hydroxide, which further shows that the carbonization is incomplete at low temperatures. This phenomenon is consistent with the corresponding SEM image. When the carbonization temperatures are 40 °C, 60 °C, and 80 °C, the carbonized products are all Y_2_(CO_3_)_3_·2H_2_O (ICDD card No 24-1419) [30]. As the carbonization temperature changes, the diffraction peak intensity of yttrium carbonate crystals changes, indicating that the crystal form changes. When the temperature increases from 40 °C to 60 °C, the diffraction peak of the (020) crystal plane of the yttrium carbonate crystal does not change significantly, but the diffraction peaks of the other crystal planes are clearly enhanced, such as the (002), (101), (103), and (121) planes. This indicates that the growth rate of each crystal plane of yttrium carbonate changes and undergoes two-dimensional growth, consistent with the phenomenon observed in the SEM analysis. As the temperature increases to 80 °C, the characteristic peak of yttrium carbonate is clearly weakened, and miscellaneous peaks appear at angles of 10.67° and 28.66°, indicating that the product is not completely carbonized and converted into yttrium carbonate, which confirms the results of the abovementioned SEM analysis.

#### 3.2.2. Effects of the CO_2_ Flow Rate on the Morphology and Phase Type of Carbonization Products

The flow rate of CO_2_ directly affects the dissolution, absorption, and rate of carbonization of carbon dioxide, thereby changing the supersaturation of the system and affecting the morphology of yttrium carbonate. Thus, the effects of the CO_2_ flow rate on the morphology and phase type of carbonization products were investigated, and the results are shown in Figure 5 and Figure 6. Figure 5 shows the SEM images of carbonized products generated at different CO_2_ flow rates. The CO_2_ flow rate has a small effect on the morphology of the carbonization products. The morphologies of carbonized products obtained at different carbon dioxide flow rates are all divergent needle-like interlaced structures. Figure 6 shows the XRD patterns of the carbonization products. The phase compositions of the carbonization products are all tengerite-type yttrium carbonate, which is not obviously altered [31]. This finding may be because the dissolution of CO_2_ reaches the upper limit of saturation in the middle and late stages of carbonization. The main step for controlling the reaction rate is the dissolution of yttrium hydroxide [26]. Unilaterally changing the CO_2_ flow rate has little effect on the supersaturation of the system, and thus the morphology of the carbonization product does not change significantly.

#### 3.2.3. Effects of the Y(OH)_3_ Slurry Concentration on the Morphology and Phase Type of Carbonization Products

The concentration of the Y(OH)_3_ slurry exerts a certain effect on the concentration of Y^3+^ in the slurry and the carbonization process. The carbonization of yttrium hydroxide slurries with concentrations of 0.05, 0.10, 0.15, and 0.25 mol/L was conducted to explore the effect of the concentration of the yttrium hydroxide slurry on the morphology and phase type of the carbonization product, and samples were collected and tested after carbonization for 90 min. The results are shown in Figure 7 and Figure 8.

Figure 7 shows the SEM images of carbonized products generated with different yttrium hydroxide slurry concentrations. Figure 7 shows that the morphology of the carbonization product changes slightly with the change in slurry concentration. When the concentrations are 0.05 and 0.10 mol/L, the crystal grains of the carbonization product are larger, the aspect ratio is relatively small, and the carbonization product exhibits a long flaky shape. When the slurry concentration increases to 0.15 and 0.25 mol/L, the crystal grains of the carbonization product become smaller and have a needle-like structure and a relatively large aspect ratio. When the concentration is low, the rate at which yttrium hydroxide dissolves and the release of Y^3+^ are low, and the concentration of Y^3+^ is low. Carbonization at the same CO_2_ flow rate results in low supersaturation of yttrium carbonate and a slow grain growth rate, which is conducive to the full growth of each crystal face of the crystal grains; the crystal grains of the obtained carbonization product are larger. When the slurry concentration is high, high supersaturation occurs, and the crystal grains of the obtained carbonization product are small. Typically, when the concentration of the reactants is high, the reaction speed is faster, but the precipitated particles obtained from a reaction solution with an excessively high concentration are smaller and easily agglomerate, forming a large, agglomerated carbonization product, as shown in Figure 7d, and the dispersibility is poor. Moreover, increasing the concentration of the yttrium hydroxide slurry will further reduce the fluidity of the carbonization system, resulting in poor carbonization efficiency. Figure 8 shows the XRD patterns of the carbonization products, and all of the samples, formed at different YCl_3_ concentrations, are yttrium carbonate with no peaks attributed to impurities [30].

#### 3.2.4. Effects of the Stirring Speed on the Morphology and Phase Type of Carbonization Products

Stirring is one of the common methods to accelerate a chemical reaction rate. However, for gas–liquid–solid reaction systems, stirring exerts more complicated effects on gas absorption and dissociation and the dissolution of yttrium hydroxide. Stirring alters the degree of mixing of the system, thereby affecting the crystallization of carbonized products. The carbonization experiments were performed at stirring speeds of 300 rpm, 400 rpm, 500 rpm, 700 rpm, and 900 rpm to explore the effects of the stirring speed on the morphology and phase type of the carbonized product. Figure 9 shows the SEM images of carbonization products generated at different stirring speeds. When the stirring speed is 300 rpm, the carbonized product has a sheet-like structure formed by the needle assembly, and the crystal grains are larger. As the stirring speed increases above 400 rpm, the morphology of the carbonized product returns to the original needle-like morphology with smaller crystal grains. This change is attributed to the increased collision probability of the stirrer and the carbonized product crystals when the stirring speed increases, and crystal grain breakage occurs. On the other hand, since the reaction rate in the middle and late stages of carbonization is controlled by the dissolution of yttrium hydroxide, a higher stirring rate can increase the diffusion speed, accelerate the carbonization reaction, increase the relative supersaturation of the system, and achieve a nucleation rate that is greater than the crystal growth rate, which is conducive to the preparation of crystals with smaller crystal grains. Typically, changing the stirring speed has little effect on the morphology of carbonization products. Figure 10 shows the XRD patterns of the carbonization products obtained at different stirring speeds. The XRD results in Figure 10 show that the carbonization products are all yttrium carbonate hydrate, and no significant change in the phase composition is observed.

### 3.3. Research on the Crystallization Mechanism of Needle-like Yttrium Carbonate

Through the analysis of the effects of carbonization conditions on the morphology of yttrium carbonate described above, temperature had a greater effect on the morphology of yttrium carbonate than the other experimental parameters. When the carbonization temperature is fixed, for example, at 40 °C, the CO_2_ flow rate, the concentration of the yttrium hydroxide slurry, and the stirring speed have little effect on the morphology and phase type of the resulting yttrium carbonate. The morphology of all yttrium carbonate products is needle-like aggregates, and the phases are all tengerite-type yttrium carbonate. An experiment was conducted at a reaction temperature of 40 °C, a yttrium hydroxide slurry concentration of 0.15 mol/L, a CO_2_ flow rate of 0.10 L/min, and a stirring speed of 700 rpm to explore the mechanism underlying the morphological evolution of acicular yttrium carbonate. The carbonization products obtained at different times were analyzed using XRD, SEM, and TEM to obtain the carbonized crystal growth mechanism and provide theoretical guidance for carbonization to synthesize needle-like yttrium carbonate crystals.

#### 3.3.1. XRD, SEM, and TEM Analyses of Changes in the Carbonization Products Formed at Different Carbonization Times

The phase and morphology of samples obtained at different carbonization times were analyzed to determine the crystal growth process of carbonized products during the carbonization process, and the results are shown in Figure 11, Figure 12 and Figure 13.

Figure 11 shows the XRD pattern of the carbonized product generated during the carbonization process. As shown in Figure 11, when the carbonization time is less than 30 min, the carbonized product has a wider diffraction peak that is close to the diffraction peak of the initial yttrium hydroxide and poor crystallinity. However, with the extension of carbonization time, the intensity of the diffraction peaks is slightly weakened, indicating that a large amount of yttrium hydroxide is not carbonized at this time. When the carbonization time is 60 min, the characteristic peak of yttrium carbonate begins to appear in the XRD pattern, but the peak intensity is weak. Carbonization continued until t = 90 min, and the carbonization product was completely converted into crystalline yttrium carbonate. By comparing the XRD standard (JCPDF#24-1419) and the test results, the carbonized product is tengerite-type carbonate yttrium. Among them, the diffraction peak corresponding to the (020) crystal plane is relatively strong, clearly indicating that oriented growth occurs in the (010) direction, which is consistent with the structure of tengerite-type yttrium carbonate that easily grows in a one-dimensional orientation, as reported in the literature [32].

SEM and TEM measurements were performed for the carbonized products generated at different times to further clarify the growth process of acicular yttrium carbonate crystals, and the results are shown in Figure 12 and Figure 13, respectively. Figure 12a shows that, when the carbonization time is t = 10 min, the morphology of the carbonized product is similar to that of yttrium hydroxide. The only difference is the formation of a small amount of needle-like products on the surface of the carbonization product after 10 min, and its surface is fluffier than that of yttrium hydroxide. As shown in the TEM image presented in Figure 13a, a small amount of needle-like products is mixed with the sheet-like yttrium hydroxide. At the initial stage of carbonization, the slurry system is weakly alkaline, and CO_2_ continuously dissolves into the liquid phase and rapidly reacts with OH^−^ ionized by yttrium hydroxide to form HCO_3_^−^, which promotes a decrease in the pH of the system and the dissolution of aggregates of yttrium hydroxide nanoplatelets. Then, the Y^3+^ released by the dissolution of yttrium hydroxide encounters a large amount of HCO_3_^−^ on the surface, increasing the supersaturation of yttrium carbonate and causing it to nucleate and grow heterogeneously on the surface of the yttrium hydroxide. After carbonization for t = 30 min, the yttrium hydroxide nanosheet aggregates further dissolved and collapsed, and the needle-like products appeared and the degree of fluffiness on the surface became increasingly obvious, as shown in Figure 12b. When the carbonization reaches t = 60 min, most of the yttrium hydroxide has been dissolved, more acicular yttrium carbonate nucleates and grows heterogeneously on the surface of the yttrium hydroxide, and characteristic peaks are observed for yttrium carbonate in the XRD pattern, as shown in Figure 12c. As shown in Figure 12, compared to the SEM image of the carbonized product at 10 min, the amount of flaky yttrium hydroxide produced after 60 min of carbonization is significantly reduced, while the needle-like products are increasing, which are also observed in the TEM image shown in Figure 13b. After carbonization with CO_2_ for t = 90 min, yttrium hydroxide was completely dissolved and carbonized to form needle-like yttrium carbonate, and its morphology was completely transformed into a needle-like stacked structure, as shown in Figure 11 and Figure 12d. We determined the structure of the nanoneedles by Pixelmator Pro in the TEM image. As shown in the TEM image in Figure 13c, yttrium carbonate is a needle-like crystal with a length of 1–2 µm and a width of 20–30 nm. As illustrated in the SAED and the high-magnification lattice fringe photos shown in Figure 13d, yttrium carbonate adopts a polycrystalline form with a lattice spacing of approximately 0.3 nm, corresponding to the (200) crystal plane of yttrium carbonate structure. At the same time, the high-magnification lattice fringe photos show multiple regular lattice fringe regions with a size of approximately 5–10 nm, indicating that acicular yttrium carbonate is composed of yttrium carbonate nanocrystalline grains with a size of approximately 5–10 nm. The change process is consistent with the XRD results shown in Figure 11.

#### 3.3.2. Crystal Growth Mechanism during Carbonization

Based on the aforementioned analysis, this study proposes the mechanism for the formation of acicular yttrium carbonate, as shown in Figure 14. In the carbonization process mediated by carbon dioxide, carbon dioxide is gradually absorbed and dissociates into HCO_3_^−^ and H^+^, and H^+^ reacts with OH^−^ in the slurry to form H_2_O, which promotes the dissolution of yttrium hydroxide and release of Y^3+^. When the concentrations of HCO_3_^−^ and Y^3+^ reach the Ksp of yttrium carbonate, yttrium carbonate gradually nucleates and then adheres to the surface of yttrium hydroxide for heterogeneous growth. Under normal circumstances, the Ostwald ripening (OR) mechanism refers to the growth of crystals through atomic or ion deposition, and their crystal lattices are often relatively complete [33]. However, as shown in the TEM image presented in Figure 13d, the yttrium carbonate nanoneedles are composed of multiple regular lattice fringe regions with a size of approximately 5–10 nm, indicating the presence of many defects in the crystal, which is difficult to explain using the traditional ion deposition growth mechanism.

Based on this information, this article proposed that the oriented attachment (OA) mechanism plays a dominant role in the formation of yttrium carbonate crystals. Compared with the classical growth mechanism, the OA mechanism is characterized by multiple small nanoparticles with the same crystal lattice orientation that form a new crystal with a high specific surface area, large porosity, and high crystallinity through direct bonding [34]. As carbonization proceeds, the pH value decreases, yttrium hydroxide dissolves, and yttrium carbonate nucleates to form nanocrystals. These nanocrystals tend to line up side-by-side, spontaneously fuse together through directional attachment, and finally, complete particle interface fusion through the OR mechanism to form a long needle-shaped yttrium carbonate with a flat surface. The nucleation and attachment of nanoparticles proceed very quickly, and the process is difficult to observe. In the process of nanocrystal collision and attachment, due to imperfect attachment, the dislodgement of attached crystals occurs during crystal growth, causing some defects, such as dislocations, to be introduced into the needle-shaped crystal. Driven by the OA mechanism, the morphology, size, and microstructural evolution of the material will show characteristics different from traditional crystals grown by dissolution and deposition, forming the lattice fringe phase diagram shown in Figure 13d.

### 3.4. Characterization of Yttrium Oxide Nanoneedle Powder

The yttrium carbonate crystal obtained through carbonization was calcined at 600 °C for 2 h to obtain the acicular yttrium oxide powder, and the calcined product was analyzed using XRD and SEM. Figure 15 shows the XRD pattern of the calcined product, which is consistent with the standard card (PDF#41-1105), and the diffraction peaks are relatively sharp and strong with almost no peaks attributed to impurities, indicating that the calcined product consists of complete crystals with high purity. Thus, yttrium carbonate synthesized by carbonization can be calcined at 600 °C for 2 h to obtain cubic yttrium oxide with a high crystallinity. Figure 16 shows the SEM and TEM images of yttrium oxide. Compared with the SEM images of yttrium carbonate obtained using carbonization, yttrium oxide better inherited the nanoneedle morphology of yttrium carbonate, which is a polycrystalline structure with a length of 1–2 µm and a width of 20–30 nm. The CO_2_ carbonization method has the advantages of no ammonia nitrogen pollution, no need for additives, low cost of raw materials, and easy large-scale production. However, the carbonization method also has drawbacks. For example, due to the use of sodium hydroxide to adjust pH, sodium chloride high-salt wastewater will be generated; at the same time, the reaction involves a gas–liquid–solid three-phase reaction, and the operation control needs to be more refined.

## 4. Conclusions

The carbonization reaction of the alkaline system was conducted in a gas–liquid–solid-phase reaction system. The complete process includes CO_2_ dissolution and absorption, dissolution of yttrium hydroxide, and crystallization of yttrium carbonate. During the carbonization process, carbon dioxide gradually is absorbed and dissociates into HCO_3_^−^ and H^+^, and then H^+^ reacts with OH^−^ in the slurry to form H_2_O, promoting the dissolution of yttrium hydroxide and the release of Y^3+^. When the concentrations of HCO_3_^−^ and Y^3+^ in the solution reach supersaturation with respect to yttrium carbonate, yttrium carbonate begins to nucleate and grow.

The morphology and phase type of the carbonized product are substantially affected by temperature. At 20 °C, due to the low temperature, the carbonization reaction speed is low, the carbonization process is incomplete, and the morphology of yttrium hydroxide is still retained. When the temperature is higher than 40 °C, the product transforms into yttrium carbonate, and its morphology shifts from one-dimensional growth to two-dimensional growth. Conditions, such as the CO_2_ flow rate, stirring speed, and feed solution concentration, have little effect on the morphology because the lower free energy crystal face (020) of the tengerite-type yttrium carbonate easily forms needle-like crystals. When the temperature is constant, the carbonization process has universal applicability and is suitable for industrial production.

During the process of growing yttrium carbonate crystals, the nanocrystals tend to be arranged side-by-side and spontaneously fuse through directional attachment and complete particle interface fusion mediated by the OR mechanism to ultimately form flat-surfaced nanoneedle yttrium carbonate with a length of 1–2 µm and a width of 20–30 nm. The acicular yttrium carbonate was calcined in a 600 °C muffle furnace for 2 h to obtain a 20–30 nm wide acicular polycrystalline yttrium oxide powder with good morphology.

## Figures and Tables

**Figure 1 nanomaterials-12-03440-f001:**
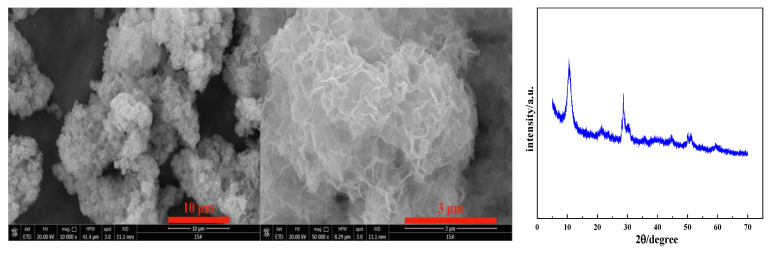
SEM and XRD patterns of Y(OH)_3_.

**Figure 2 nanomaterials-12-03440-f002:**
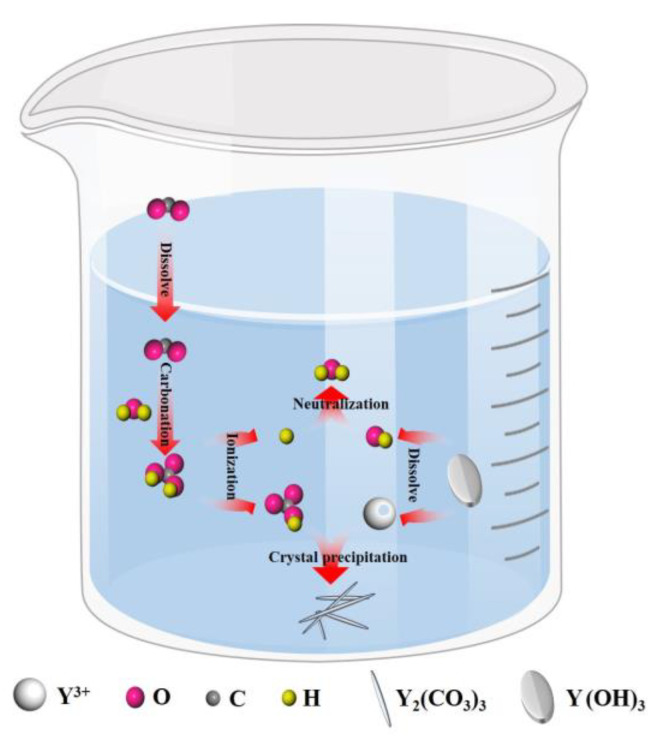
Schematic diagram of the process of preparing yttrium oxide by carbon dioxide carbonization in an alkaline system.

**Figure 3 nanomaterials-12-03440-f003:**
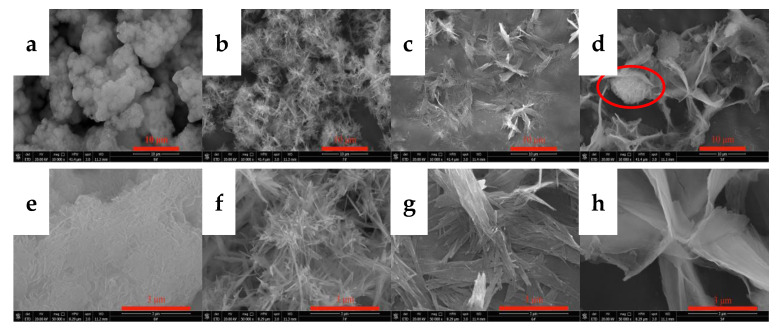
SEM images of carbonized products generated at different reaction temperatures. (Y(OH)_3_ concentration: 0.15 mol/L; carbonization stirring speed: 700 rpm; CO_2_ flow rate: 0.1 L/min). (**a**,**e**): 20 °C; (**b**,**f**): 40 °C; (**c**,**g**): 60 °C; and (**d**,**h**): 80 °C.

**Figure 4 nanomaterials-12-03440-f004:**
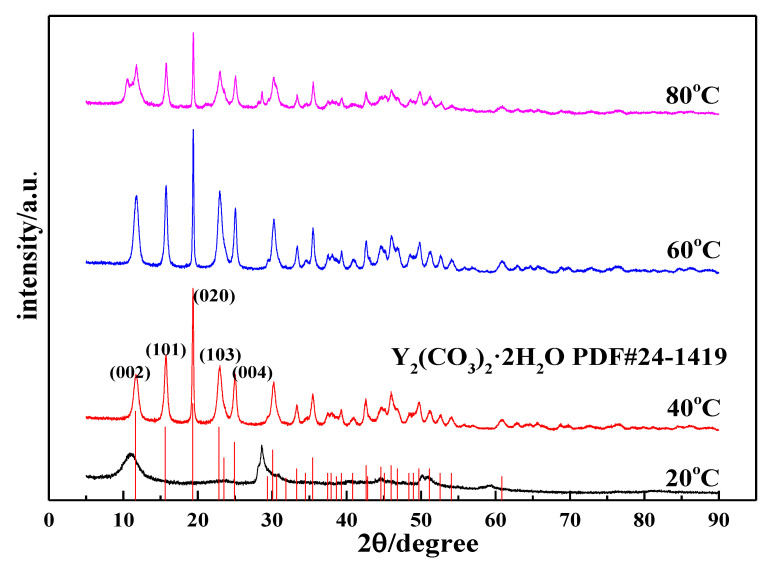
XRD patterns of carbonized products at different temperatures (Y(OH)_3_ concentration: 0.15 mol/L; carbonization stirring speed: 700 rpm; CO_2_ flow rate: 0.1 L/min).

**Figure 5 nanomaterials-12-03440-f005:**
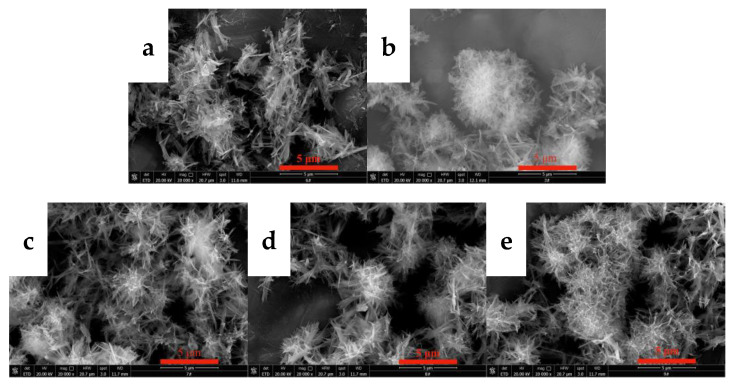
SEM images of carbonized products at different CO_2_ flow rates (Y(OH)_3_ concentration: 0.15 mol/L; carbonization stirring speed: 700 rpm; carbonization temperature: 40 °C). (**a**): 0.06 L/min; (**b**): 0.1 L/min; (**c**): 0.2 L/min; (**d**): 0.4 L/min; and (**e**): 0. 6 L/min.

**Figure 6 nanomaterials-12-03440-f006:**
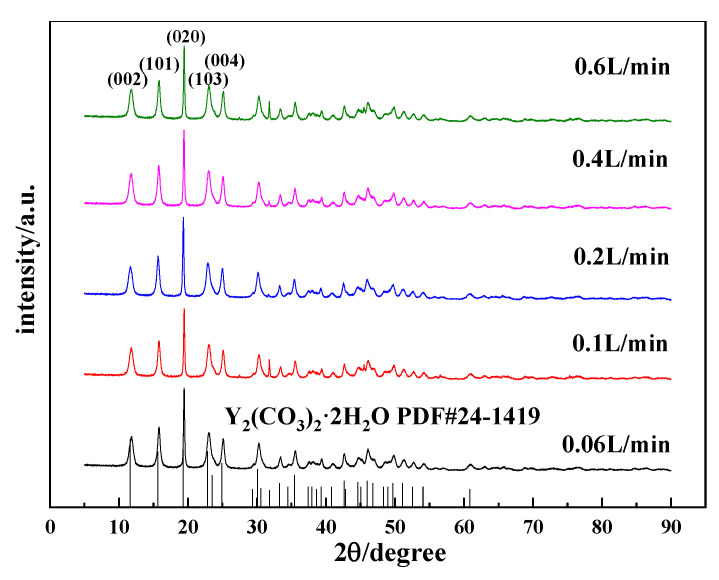
XRD patterns of carbonized products generated at different CO_2_ flow rates (Y(OH)_3_ concentration: 0.15 mol/L; carbonization stirring speed: 700 rpm; carbonization temperature: 40 °C).

**Figure 7 nanomaterials-12-03440-f007:**
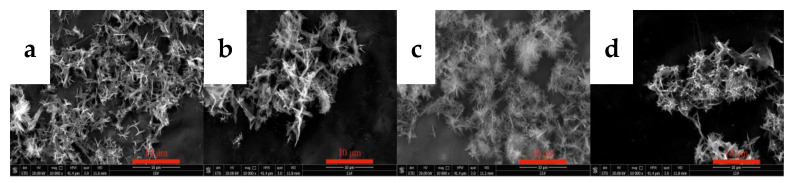
SEM images of carbonized products generated at different yttrium hydroxide slurry concentrations (carbonization temperature: 40 °C; carbonization stirring speed: 700 rpm; carbon dioxide flow rate: 0.1 L/min) (**a**): 0.05 mol/L; (**b**): 0.1 mol/L; (**c**): 0.15 mol/L and (**d**): 0.25 mol/L.

**Figure 8 nanomaterials-12-03440-f008:**
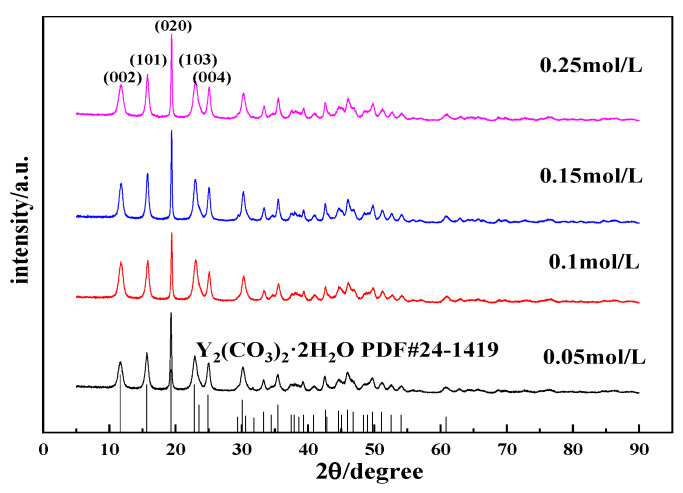
XRD patterns of carbonized products generated with different yttrium hydroxide slurry concentrations (carbonization temperature: 40 °C; carbonization stirring speed: 700 rpm; carbon dioxide flow rate: 0.1 L/min).

**Figure 9 nanomaterials-12-03440-f009:**
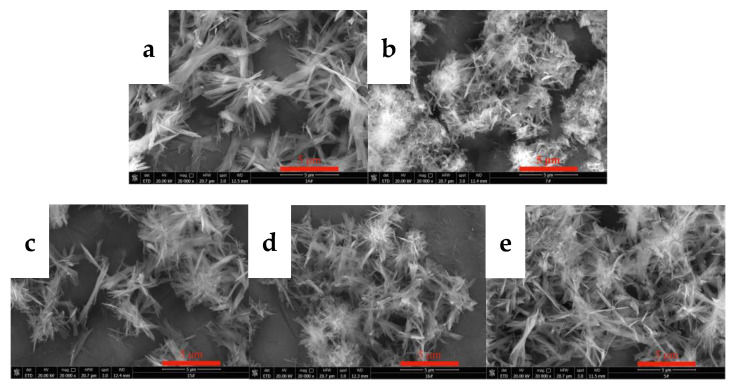
SEM images of carbonized products obtained at different stirring speeds (carbonization temperature: 40 °C, yttrium hydroxide concentration: 0.15 mol/L; carbon dioxide flow rate: 0.1 L/min). (**a**): 300 rpm; (**b**): 400 rpm; (**c**): 500 rpm; (**d**): 700 rpm; and (**e**): 900 rpm).

**Figure 10 nanomaterials-12-03440-f010:**
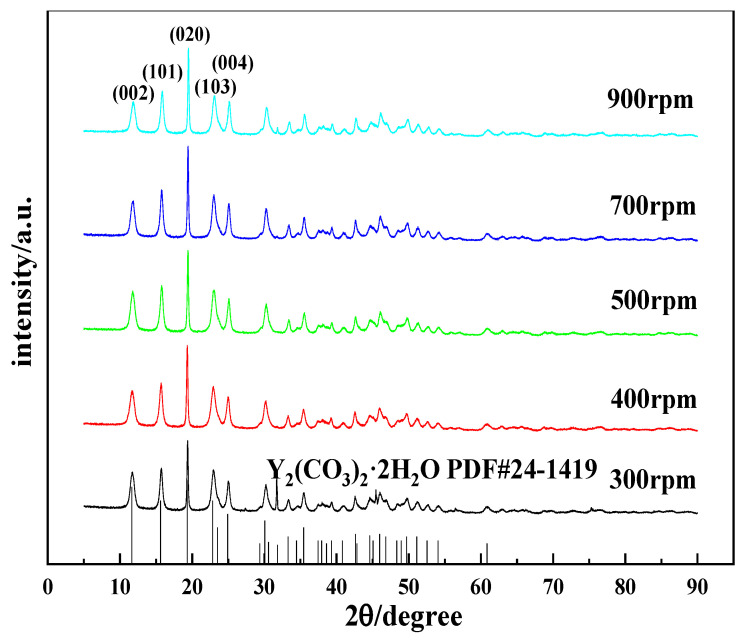
XRD patterns of carbonized products obtained at different stirring speeds (carbonization temperature: 40 °C, yttrium hydroxide concentration: 0.15 mol/L; carbon dioxide flow rate: 0.1 L/min).

**Figure 11 nanomaterials-12-03440-f011:**
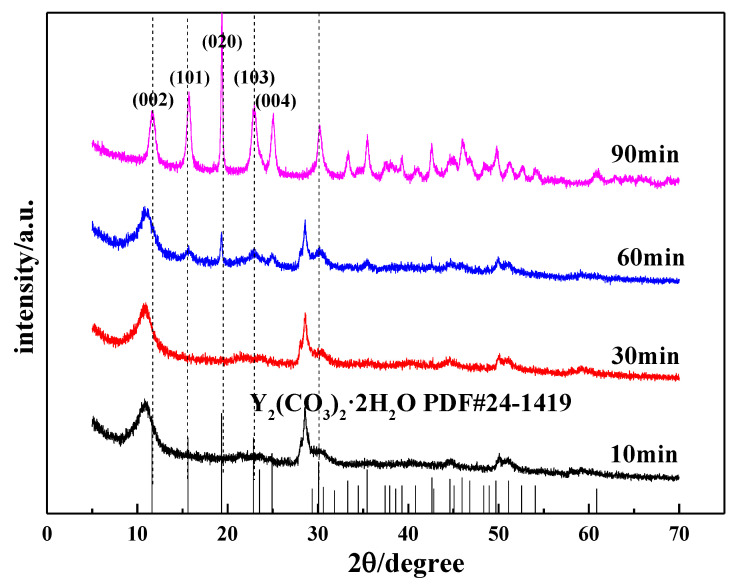
Changes in the XRD patterns of carbonized products during the carbonization process.

**Figure 12 nanomaterials-12-03440-f012:**
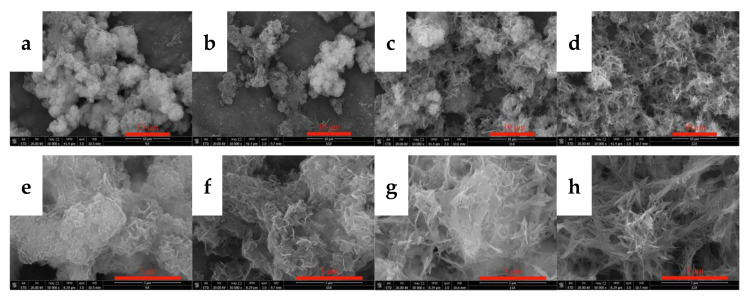
SEM images showing changes observed during the carbonization process. (**a**,**e**): 10 min; (**b**,**f**): 30 min; (**c**,**g**): 60 min; (**d**,**h**): 90 min.

**Figure 13 nanomaterials-12-03440-f013:**
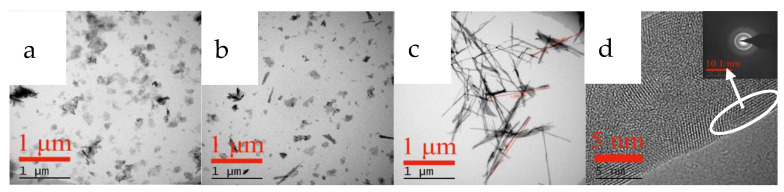
TEM images showing changes observed during the carbonization process. (**a**): 10 min; (**b**): 60 min; (**c**): 90 min; (**d**): 90 min.

**Figure 14 nanomaterials-12-03440-f014:**
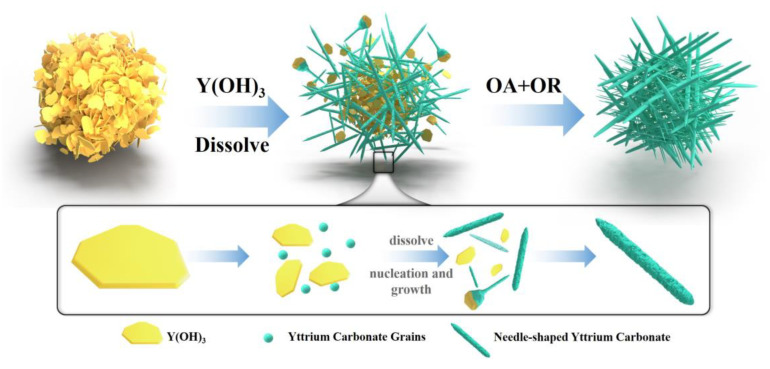
Schematic diagram of the mechanism underlying the formation of needle-shaped yttrium carbonate.

**Figure 15 nanomaterials-12-03440-f015:**
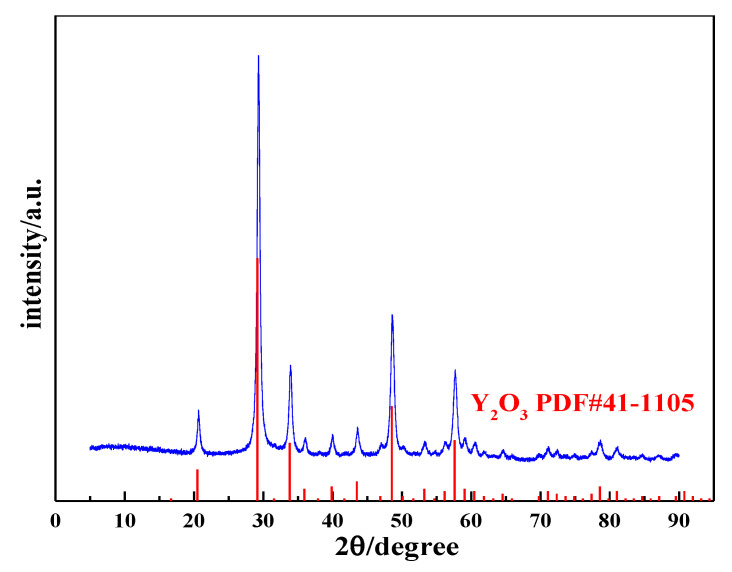
XRD pattern of yttrium oxide.

**Figure 16 nanomaterials-12-03440-f016:**
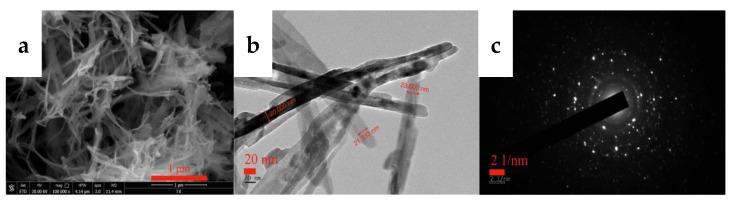
SEM (**a**) and TEM (**b**,**c**) images of yttrium oxide.

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
