# Peer review of "Synthesis of Yttrium Oxide Nanoneedles with Carbon Dioxide Carbonization"

_nanomaterials, 2022, doi:10.3390/nano12193440_

Round 1
Reviewer 1 Report
In this manuscript, Rao and co-workers introduce a CO2 carbonization synthetic method of yttrium oxide nanoneedles. The effects of temperature, CO2 flow rate and stirring speed during the synthetic process are investigated. The growth process through oriented attachment and Ostwald ripening was also demonstrated. The characterization results and data provided by the authors basically support their conclusions. Overall, this is a detailed introduction to their experiments, which may provide experience and reference for the scholars of material synthesis. In this respect, I support its publication.
However, as the authors say, this method is not newly proposed. Similar structures have been prepared by the same method (Journal of Solid State Chemistry 2020, 290, 121593; Small 2019, 15, 1902249; also reference 12, 13, 18 and 19 in this manuscript). Therefore, the authors have to seriously consider the innovativeness and importance of this work. Before major significance can be fully demonstrated, I cannot recommend its publication.
Some specific comments:
1.The authors address the advantages of this type of synthetic method in Introduction and claim that the method is cheap, facile, efficient, low-polluting, and easy to use on a large scale. It looks like this approach works flawlessly. Authors are expected to make objective evaluations of their research work. The shortcomings of such methods should also be properly assessed.
2.The quality of all SEM and TEM images needs to be greatly improved. The scale bar in the TEM and SEM images should be much clearer.
3.The size distribution of nanoneedles should be provided.
Reviewer 2 Report
My comments are in attached file.

Author Response
The revised material has been submitted in the attachment

Round 2
Reviewer 1 Report
The authors have revised the manuscript well. The current version is suitable for publication.